# LEMUR: INTEGRATING LARGE LANGUAGE MODELS IN AUTOMATED PROGRAM VERIFICATION

**Haoze Wu,**[*] **Clark Barrett**
Department of Computer Science
Stanford University
{haozewu, barrett}@stanford.edu

**Nina Narodytska**
VMware Research
VMware by Broadcom
n.narodytska@gmail.com

## ABSTRACT

The demonstrated code-understanding capability of LLMs raises the question of whether they can be used for automated program verification, a task that demands high-level abstract reasoning about program properties that is challenging for verification tools. We propose a general methodology to combine the power of LLMs and automated reasoners for automated program verification. We formally describe this methodology as a set of transition rules and prove its soundness. We instantiate the calculus as a sound automated verification procedure and demonstrate practical improvements on a set of synthetic and competition benchmarks.

## 1 INTRODUCTION

AI-powered language models are being routinely used to help developers. Examples include program synthesis from natural language descriptions by GPT-4 (OpenAI, 2023) or Github Copilot (Chen et al., 2021; GitHub, 2021), and repairing code (White et al., 2019), among others. These models have shown impressive results in generating correct code in many programming languages.

An important research question is whether modern AI models are capable of understanding the logic behind the programs they analyze. Recently, several approaches have been proposed to combine the strengths of formal verification and Large Language Models (LLMs) that demonstrate such capabilities. For example, Pei et al. (2023) made an important step in this direction by investigating whether LLMs can generate program properties, namely, program invariants, which remains a crucial and challenging task for automated program verification (Clarke et al., 2018). The authors demonstrated that LLMs are effective in generating program invariants on a set of synthetic Java programs. Another example is the recent work by Charalambous et al. (2023), who demonstrated that LLM models can be used to repair vulnerabilities in code, given examples of incorrect behavior. They provided compelling evidence of the complementary strengths of LLMs, which serve as a generator for code repair snippets, and formal techniques, which are used to check the correctness of the generated code. While these approaches show promise in program analysis tasks, they do not provide a formalization of the interaction between LLMs and formal verifiers, they require manual effort, or they are limited to the invariant generation process as a stand-alone procedure.

In this work, we propose a novel LLM-powered framework, LEMUR, for automated program verification tasks. Our key idea is to combine LLMs' ability to perform abstract high-level reasoning and automated reasoners' ability to perform precise low-level reasoning. Specifically, LLMs are employed to propose program invariants in the form of sub-goals, which are then checked by automated reasoners. This transforms the program verification tasks into a series of deductive steps suggested by LLMs and subsequently validated by automated reasoners. Our main contributions are:

- a novel framework for combining LLMs and automated reasoners for program verification;
- a presentation of LEMUR as a proof system and a proof of its soundness, which to the best of our knowledge, is the first formalization of such a hybrid approach;
- an instantiation of the LEMUR calculus that gives a sound and terminating algorithm;
- an implementation of the proposed framework (using OpenAI's GPT models);

---

[*]This work was mostly done during an internship at VMware Research.

- an experimental evaluation of LEMUR on two sets of benchmarks that demonstrates its efficiency compared with both existing AI-powered and conventional verification tools.

We highlight that LEMUR is the first *fully automated* framework combining LLMs and reasoners.

## 2 DEFINITIONS

Given a program $\mathcal{P}$ : Prog, a *reachability property*, or simply *property*, is a tuple $p = \langle \phi, l \rangle$, where $\phi$ : Pred is a Boolean *predicate* of the program state and $l : \mathbb{N}$ is a *program line*. The *negation* of $p$, denoted $\neg p$, is defined as $\langle \neg \phi, l \rangle$. Next we introduce several useful definitions and their properties.

**Definition 2.1.** *A property $p = \langle \phi, l \rangle$ is an* invariant *on $\mathcal{P}$, denoted* $\mathrm{Inv}(\mathcal{P}, p)$*, iff $p$ holds (i.e., $\phi$ always evaluates to true at line $l$) for all possible executions of the program $\mathcal{P}$.*

**Example 2.1.** *Consider the simple program $\mathcal{P}$ in Figure 2 (the first frame, top row). $\mathcal{P}$ instantiates an unsigned 32-bit integer variable $x$ to 0 and increases its value by 4 on each loop iteration. A property $p = \langle \phi, l \rangle$ is specified on the 4th line, so $\phi = (x \mathrel{!=} 30)$ and $l = 4$ for this property. It is easy to see that $p$ is an invariant as $x$ will never be divisible by 3, for example.* ∎

An assumption $q = \langle \phi, l \rangle$ is a condition that is assumed in a program.

**Definition 2.2.** *An* assumption $q = \langle \phi, l \rangle$ *modifies a program $\mathcal{P}$ as follows:*

1. *if $\phi$ holds at line $l$ during some execution of $\mathcal{P}$, then $\mathcal{P}$ continues execution without changes;*
2. *if $\phi$ does not hold at line $l$ during some execution of $\mathcal{P}$, then that execution terminates at $l$.*

We use $\mathcal{P}' = \mathrm{Asm}(\mathcal{P}, q)$ to denote the modification of $\mathcal{P}$ with the assumption $q$. An assumption can itself be an invariant. We now introduce a special notion of an *implication*.

**Definition 2.3.** *Let $\mathcal{P}$ be a program, and $p$, $q$ be properties on $\mathcal{P}$. We say that $q$* implies $p$ *with respect to $\mathcal{P}$, denoted $q \xrightarrow[\mathcal{P}]{} p$, iff $p$ is an invariant on $\mathrm{Asm}(\mathcal{P}, q)$.*

**Example 2.2.** *Consider the program $\mathcal{P}$ in Figure 2 and an assumption $q = \langle \phi = (x \mathrel{\%} 4 == 1), l = 3 \rangle$, with $\mathcal{P}' = \mathrm{Asm}(\mathcal{P}, q)$ (depicted in the first frame in the bottom row of Figure 2). To see the difference between $\mathcal{P}$ and $\mathcal{P}'$, observe that the loop is executed only once in $\mathcal{P}'$. This is because $x=0$ when entering the loop, so $(x \mathrel{\%} 4) \mathrel{!=} 1$, which is the formula $\phi$ in $q$, does not hold, and therefore, $\mathcal{P}'$ terminates. If we consider an alternative assumption $q' = \langle \phi = (x \mathrel{\%} 4 == 0), l = 3 \rangle$ (depicted in the fourth frame at the bottom of Figure 2), we can see that its predicate $\phi$ holds for all executions. Hence, $q'$ is an invariant for $\mathcal{P}$. Finally, we can see $q' \xrightarrow[\mathcal{P}]{} p$, where $p$ is from Example 2.1.* ∎

The following propositions follow from the definitions above.

**Proposition 2.1.** *Let $\mathcal{P}$ be a program, and $p$, $q$ be properties on $\mathcal{P}$:*

- *The property $p$ is an invariant on $\mathcal{P}$ if $q$ is an invariant on $\mathcal{P}$ and $q$ implies $p$ with respect to $\mathcal{P}$. More formally, $(\mathrm{Inv}(\mathcal{P}, q) \wedge q \xrightarrow[\mathcal{P}]{} p) \Rightarrow \mathrm{Inv}(\mathcal{P}, p)$.*
- *The property $p$ is not an invariant on $\mathcal{P}$ if the property $p$ is not an invariant on $\mathcal{P}' = \mathrm{Asm}(\mathcal{P}, q)$. More formally, $\neg\mathrm{Inv}(\mathcal{P}', p) \Rightarrow \neg\mathrm{Inv}(\mathcal{P}, p)$.*

**Proposition 2.2.** *For any property $p$ on a program $\mathcal{P}$, $p \xrightarrow[\mathcal{P}]{} p$.*

**Proposition 2.3.** *For any properties $p, q, r$ on a program $\mathcal{P}$, if $p \xrightarrow[\mathcal{P}]{} q$ and $q \xrightarrow[\mathcal{P}]{} r$, then $p \xrightarrow[\mathcal{P}]{} r$.*

Note that it is possible that neither a property $p$ nor its negation $\neg p$ is an invariant on a program.

**Example 2.3.** *Consider again our example from Example 2.1 and two properties at line 3: $p = \langle \phi = (x \mathrel{\%} 8 == 4), l = 3 \rangle$ and $p' = \langle \phi' = (x \mathrel{\%} 8 \mathrel{!=} 4), l = 3 \rangle$. Neither $p$ nor $p'$ is an invariant on $\mathcal{P}$. On the first loop iteration, we have that $x=0$ before line 3, so $\phi'$ holds and $\phi$ does not at line 3. But on the second loop iteration, we have that $x=4$ before line 3, so $\phi$ holds and $\phi'$ does not.* ∎

**Definition 2.4.** *A property $p = \langle \phi, l \rangle$ is* stable *for $\mathcal{P}$, denoted $\mathcal{S}(\mathcal{P}, p)$, if, for each execution of the program, either $\phi$ always evaluates to true at line $l$ or $\phi$ always evaluates to false at line $l$.*

**Example 2.4.** *Consider an example to illustrate the definition of stability.*

```
Line 1: uint32_t x=rand();
Line 2: assert(x==1);
```

*The property in Line 2 is stable, as it always evaluates to true or false within a single execution.* ∎

An invariant must be stable, but a property that is not an invariant might still be stable. For example, any property on a program without loops is stable. If $p$ is stable, then $\neg p$ is also stable. Lemma 2.1 exploits stable invariants (see a proof in the extended version of the paper (Wu et al., 2023)).

**Lemma 2.1.** *Consider a program $\mathcal{P}$, two properties $p$, $q$ on $\mathcal{P}$, and a program $\mathcal{P}' = \mathrm{Asm}(\mathcal{P}, q)$. The property $p$ is an invariant on $\mathcal{P}$ if: (i) $q$ is stable for $\mathcal{P}$; (ii) $q$ implies $p$ with respect to $\mathcal{P}$; and (iii) $\neg q$ implies $p$ with respect to $\mathcal{P}$. More formally: $\mathcal{S}(\mathcal{P}, q) \wedge (q \to p) \wedge (\neg q \to p) \Rightarrow \mathrm{Inv}(\mathcal{P}, p)$.* Assume we have a verifier $\mathcal{V} : \mathrm{Prog} \times \mathbb{P}(\mathrm{Prop}) \times \mathrm{Prop} \mapsto \{\text{TRUE}, \text{FALSE}, \text{UNKNOWN}\}$, which takes as inputs a program $\mathcal{P}$, a set of assumptions $\mathcal{A}$ and a property $p$, and checks whether $\mathcal{A}$ implies $p$. More precisely, given set of assumptions $\mathcal{A} = \{q_1, \ldots, q_n\}$, we construct a new program $\mathcal{P}' = \mathrm{Asm}(\mathrm{Asm}((\ldots, \mathrm{Asm}(\mathcal{P}, q_1)), q_{n-1}), q_n)$, and the verifier checks if $p$ is an invariant on this program. Hence, a statement that $\mathcal{A}$ implies $p$ on $\mathcal{P}$ means that $p$ is an invariant on $\mathcal{P}'$. We further assume that $\mathcal{V}$ is *sound*, meaning if $\mathcal{V}$ returns TRUE, then $\mathcal{A}$ implies $p$, and if $\mathcal{V}$ returns FALSE, then $\mathcal{A}$ does not imply $p$. Note that $\mathcal{A}$ can be empty, in which case the verifier just checks whether $p$ is an invariant. When the verifier $\mathcal{V}$ returns TRUE, we say $p$ is proven; and when $\mathcal{V}$ returns FALSE, we say the property is *falsified*. $\mathcal{V}$ is incomplete, meaning that $\mathcal{V}$ can return UNKNOWN.

In practice, $\mathcal{V}$ can be instantiated with an automated program verifier such as CBMC (Kroening & Tautschnig, 2014), ESBMC (Gadelha et al., 2018), or UAUTOMIZER (Heizmann et al., 2013). We provide an overview of the main techniques employed by these tools in the extended version of the paper (Wu et al., 2023) and note here that a crucial challenge shared across existing verifiers is the automatic decomposition of a verification task into smaller, more manageable sub-tasks. This decomposition requires high-level reasoning that is difficult to automate using conventional formal methods, but plausibly could be performed by LLMs, with their documented code-understanding capability. However, it is crucial to preserve soundness when LLMs are used to automatically perform this high-level reasoning in program verification tasks.

## 3 LEMUR: INTEGRATING LLMs IN PROGRAM VERIFICATION

We present LEMUR, a proof calculus that combines LLMs and automated reasoners to prove properties of programs. The calculus operates over a *configuration*, which is either one of the distinguished symbols $\{\text{SUCCESS}, \text{FAIL}\}$ or a tuple $\langle \mathcal{P}, \mathcal{A}, \mathcal{M} \rangle$, where $\mathcal{P}$ is a program, $\mathcal{A}$ is either $\varnothing$ or a singleton representing the assumption, and $\mathcal{M}$ is a list of properties referred to as *proof goals*. $\mathcal{M}$ itself is referred to as a *trail*. The last element of $\mathcal{M}$ represents the current property to prove. We use the notation $::$ to denote a concatenation of two lists. In particular, $\mathcal{M} = \mathcal{M}' :: p$ means that $\mathcal{M}$ is a concatenation of a trail $\mathcal{M}'$ and a property $p$, where $p$ is the last element of $\mathcal{M}$. The rules describe the conditions under which a certain configuration can transform into another configuration. In this calculus, deciding whether $\mathrm{Inv}(\mathcal{P}, p)$ holds reduces to finding a sequence of valid rule applications from the *starting configuration* $\langle \mathcal{P}, \varnothing, [p] \rangle$ to either SUCCESS or FAIL.

Our calculus performs oracle calls to LLMs to propose new properties and revise them. The oracle $\mathcal{O}_{\text{propose}} : \mathrm{Prog} \times \mathrm{Prop} \mapsto \mathbb{P}(\mathrm{Prop})$ proposes new properties, given a program and the current proof goal as inputs. A key hypothesis here is that LLMs are capable of generating new properties that are likely to (i) be invariants, and (ii) imply the proof goal given in a prompt. We will discuss strategies to generate prompts in Section 4. Importantly, properties generated by an LLM are treated as assumptions until we can prove that they are invariants of the original program. The oracle $\mathcal{O}_{\text{repair}}$ revises previously proposed properties, e.g., if we determine that a property $q$ previously produced by $\mathcal{O}_{\text{propose}}$ does not hold or does not imply the current proof goal. In this case, we request an LLM to repair $q$. We have $\mathcal{O}_{\text{repair}} : \mathrm{Prog} \times \mathrm{Prop} \times \mathrm{Prop} \times \{\text{FALSE}, \text{UNKNOWN}\} \mapsto \mathbb{P}(\mathrm{Prop})$, whose inputs comprise a program, two properties, and a solver return value. The first property is the current proof goal, and the second property $q$ is an assumption previously proposed by oracles. The output of $\mathcal{O}_{\text{repair}}$ is a new set of properties. In practice, we implement it with a prompt to an LLM to either correct or strengthen $q$ (see Section 4). Finally, the calculus performs an external call to a verifier $\mathcal{V}$ to check whether a property holds.

The proof rules of LEMUR are shown in Fig. 1. Each rule defines a set of conditions that must hold for the rule to be applicable. Note again that these conditions permit invocations of LLMs and/or verifiers. The rules within the calculus can be partitioned into four groups.

The first group contains rules that are responsible for generating new proof goals for specific configurations. These rules are **Propose**, **Repair 1**, and **Repair 2**. The **Propose** rule states that if the

$$\frac{\mathcal{M} = \mathcal{M}' :: p \quad \mathcal{V}(\mathcal{P}, \mathcal{A}, p) = \textsc{Unknown} \quad q \in \mathcal{O}_{\text{propose}}(\mathcal{P}, p)}{\mathcal{P}, \mathcal{A}, \mathcal{M} \Longrightarrow \mathcal{P}, \{q\}, \mathcal{M}} \text{ (\textbf{Propose})}$$

$$\frac{\mathcal{A} = \{q\} \quad \mathcal{M} = \mathcal{M}' :: p \quad \mathcal{V}(\mathcal{P}, \mathcal{A}, p) = \textsc{True}}{\mathcal{P}, \mathcal{A}, \mathcal{M} \Longrightarrow \mathcal{P}, \varnothing, \mathcal{M} :: q} \text{ (\textbf{Decide})}$$

$$\frac{\mathcal{M} = \mathcal{M}' :: p :: q \quad \mathcal{V}(\mathcal{P}, \mathcal{A}, q) \neq \textsc{True} \quad q' \in \mathcal{O}_{\text{propose}}(\mathcal{P}, p)}{\mathcal{P}, \mathcal{A}, \mathcal{M} \Longrightarrow \mathcal{P}, \{q'\}, \mathcal{M}' :: p} \text{ (\textbf{Backtrack})}$$

$$\frac{\mathcal{A} = \{q\} \quad \mathcal{M} = \mathcal{M}' :: p \quad \mathcal{V}(\mathcal{P}, \mathcal{A}, p) = \textsc{Unknown} \quad q' \in \mathcal{O}_{\text{repair}}(\mathcal{P}, p, q, \textsc{Unknown})}{\mathcal{P}, \mathcal{A}, \mathcal{M} \Longrightarrow \mathcal{P}, \{q'\}, \mathcal{M}' :: p} \text{ (\textbf{Repair 1})}$$

$$\frac{\mathcal{A} = \varnothing \quad \mathcal{M} = \mathcal{M}' :: p :: q \quad \mathcal{V}(\mathcal{P}, \mathcal{A}, q) = \textsc{False} \quad q' \in \mathcal{O}_{\text{repair}}(\mathcal{P}, p, q, \textsc{False})}{\mathcal{P}, \mathcal{A}, \mathcal{M} \Longrightarrow \mathcal{P}, \{q'\}, \mathcal{M}' :: p} \text{ (\textbf{Repair 2})}$$

$$\frac{\mathcal{A} = \varnothing \quad \mathcal{M} = \mathcal{M}' :: p \quad \mathcal{V}(\mathcal{P}, \mathcal{A}, p) = \textsc{True}}{\mathcal{P}, \mathcal{A}, \mathcal{M} \Longrightarrow \textsc{success}} \text{ (\textbf{Success 1})}$$

$$\frac{\mathcal{A} = \varnothing \quad \mathcal{M} = \mathcal{M}' :: p :: q \quad \mathcal{S}(\mathcal{P}, q) \quad \mathcal{V}(\mathcal{P}, \{\neg q\}, p) = \textsc{True}}{\mathcal{P}, \mathcal{A}, \mathcal{M} \Longrightarrow \textsc{success}} \text{ (\textbf{Success 2})}$$

$$\frac{\mathcal{M} = [p] \quad \mathcal{V}(\mathcal{P}, \mathcal{A}, p) = \textsc{False}}{\mathcal{P}, \mathcal{A}, \mathcal{M} \Longrightarrow \textsc{fail}} \text{ (\textbf{Fail})}$$

Figure 1: Deductive rules of the LEMUR calculus.

verifier is unable to prove or disprove the current proof goal $p$, we can invoke the $\mathcal{O}_{\text{propose}}$ oracle to obtain a property $q$ and update $\mathcal{A}$ to be $\{q\}$. The **Repair 1** rule can be applied when the current assumption $q$ is not sufficient for the verifier to prove the current proof goal $p$. In this case, we can use the oracle $\mathcal{O}_{\text{repair}}$ to propose ways to strengthen $q$ and choose one of them, $q'$, as the new assumption. The **Repair 2** rule can also be applied when $q$ is already in the trail but is falsified by the verifier $\mathcal{V}$. In this case, we can use $\mathcal{O}_{\text{repair}}$ to repair $q$ and update $\mathcal{A}$ accordingly.

The second group contains only the **Decide** rule. This rule allows an assumption $q$ to become a new goal (i.e., be appended to $\mathcal{M}$) if the verifier $\mathcal{V}$ is able to prove that $q$ implies the current goal.

The third group allows LEMUR to recover from faulty assumptions. The **Backtrack** rule is applicable when there are at least two elements in the trail and the verifier cannot prove the current proof goal. It allows LEMUR to revert to the previous proof goal (the second to last property in the trail $\mathcal{M}$) and pick a different assumption suggested by $\mathcal{O}_{\text{propose}}$. Note that **Backtrack** might not be the only applicable rule. For example, **Repair 1** or **Repair 2** could also be applicable. In practice, we use a strategy to decide between multiple applicable rules. This is discussed in Sec. 4.

The final group specifies three termination conditions for the calculus. The **Success 1** rule states that whenever the assumption is empty and the verifier is able to prove the current proof goal (i.e., the last property $p$ in the trail $\mathcal{M}$), we can transition into the SUCCESS state. Note that if the verifier can prove the original property, then this rule can be directly applied to the starting configuration to reach SUCCESS. **Success 2** states that if the last two elements of the trail $\mathcal{M}$ are $p$ and $q$, the current proof goal $q := \langle \phi, l \rangle$ is stable (as defined in Sec. 2), and the verifier is also able to also prove $p$ under the assumption $\langle \neg \phi, l \rangle$, then $p$ is an invariant (as a consequence of Lemma 2.1), and we can transition to SUCCESS. The **Success 2** rule constitutes a way to utilize an *incorrect sub-goal* $q$ proposed by the LLM-based oracles to decompose the verification task: we separately reason about the cases when $q$ holds and when it does not hold. Finally, if the verifier $\mathcal{V}$ proves that the original property is not an invariant, whether under an assumption or not, then we transition to the FAIL state using **Fail**.

Note that the program $\mathcal{P}$ remains unchanged in every rule. We keep it as part of the state for two reasons. First, it is convenient for keeping the calculus self-contained, as $\mathcal{P}$ is an input to the verifiers and the oracles. Second, in the future, it might be possible to augment the calculus with rules that update $\mathcal{P}$, by, for example, rewriting the program using LLMs in an invariant-preserving manner.

We state the following soundness properties about LEMUR. The proof is presented in the extended version of the paper (Wu et al., 2023).

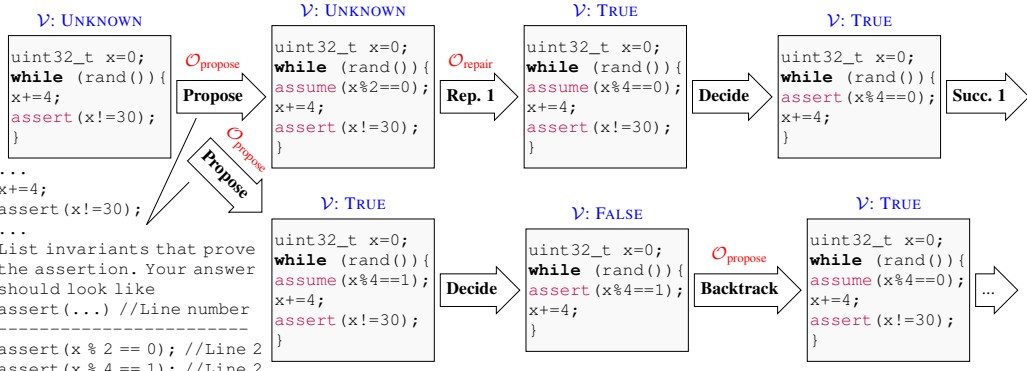

Figure 2: A running example of executing the LEMUR calculus.

**Theorem 3.1** (Soundness). *Given a property $p_0$ and a program $\mathcal{P}$, if* SUCCESS *is reached by a sequence of valid rule applications starting from $\langle \mathcal{P}, \varnothing, [p_0] \rangle$, then $p$ is an invariant on $\mathcal{P}$.*

**Theorem 3.2** (Soundness 2). *Given a property $p_0$ and a program $\mathcal{P}$, if* FAIL *is reached by a sequence of valid rule applications starting from $\langle \mathcal{P}, \varnothing, [p_0] \rangle$, then $p$ is not an invariant on $\mathcal{P}$.*

***Example* 3.1.** *To provide more intuition about the proof system and to motivate the design choices when instantiating* LEMUR, *we consider again our running example. Figure 2 illustrates how* LEMUR *can be used to verify properties in practice. In Figure 2 each frame represents a state of the program. Transitions between states are depicted by arrows, with each arrow marked with the rule applied to execute this transition. In this example, our goal is to prove the property* x!=30 *in a while loop that keeps adding 4 to an unsigned 32-bit integer variable* x. *We note that this particular verification task is adapted from a similar one in the SV-COMP competition.[1] While seemingly trivial, during the competition, 19 out of the 24 participating tools (including the overall winner of the competition* UAUTOMIZER*) were not able to solve this benchmark.*

*The initial configuration is* $\langle \mathcal{P}, \varnothing, [p] \rangle$, *where* $\mathcal{P}$ *is the given program and* $p = \langle$ x!=30, 3$\rangle$.[2] *Suppose the verifier* $\mathcal{V}$ *is unable to solve this problem and returns* UNKNOWN. *In this case, we need to generate a new proof goal, so the only rule we can apply is* **Propose***. To do so, we invoke the LLM-based oracle* $\mathcal{O}_{propose}$ *to obtain a set of new properties that are potentially themselves invariants and might help prove the property. An example prompt is given in the bottom left of Figure 2. Note that this is not the exact prompt we found most useful, but we defer a full discussion of prompts and prompting strategies to Sec. 4. Suppose the oracle returns two potential predicates, both of which should hold at the beginning of the while loop (line 3):* x%2==0 *and* x%4==1. *The* **Propose** *rule allows us to make one of them the current assumption.*

Case (x%2==0): *The top row illustrates what happens when we transition to* $\langle \mathcal{P}, \{q = \langle$x%2==0, 3$\rangle\}, [p] \rangle$. *While* q *is indeed an invariant, it does not help to prove the assertion, and* $\mathcal{V}$ *still returns* UNKNOWN. *This means that the* **Repair 1** *rule is applicable, which invokes the oracle* $\mathcal{O}_{repair}$ *to strengthen* q. *Suppose in this case, the oracle suggests the predicate* $q' = $ x%4==0, *which clearly implies the original property* x!=30. *Now, suppose* $\mathcal{V}(\mathcal{P}, \{q'\}, p)$ *returns* TRUE. *We can apply the* **Decide** *rule and transition to* $\langle \mathcal{P}, \varnothing, [p, q'] \rangle$, *making* q' *the current proof goal. Proving* q' *is arguably easier because* x%4==0 *is* inductive *(i.e., if it holds in one iteration, then it will hold in the next iteration), making conventional automated reasoning techniques such as k-induction applicable. Now, if* $\mathcal{V}(\mathcal{P}, \varnothing, q') = $ TRUE, *we can apply* **Success 1** *and transition to the* SUCCESS *state, thus completing the proof.*

*We discuss the case* x%4==1 *in the extended version of the paper (Wu et al., 2023).* ∎

---

[1] https://sv-comp.sosy-lab.org/2023/results/results-verified/META_ReachSafety.table.html#/table?filter=id_any(value(jain_5-2))

[2] 3 is the line number (in the snippet) where the predicate is asserted.

---

**Algorithm 1** The LEMUR procedure

---

1: **Input:** A program $\mathcal{P}$, a property $p$.
2: **Output:** SUCCESS only if $\text{Inv}(\mathcal{P}, p)$; FAIL only if $\neg\text{Inv}(\mathcal{P}, p)$; and UNKNOWN if inconclusive.
3: **Parameters:** Verifier $\mathcal{V}$, oracles $\mathcal{O}_{\text{propose}}$ and $\mathcal{O}_{\text{repair}}$ (which satisfy Condition 1), number of proposals **k**
4: **function** lemur_check($\mathcal{P}, p$)
5:    $d \mapsto \mathcal{V}(\mathcal{P}, \varnothing, p)$
6:    **if** $d = \text{FALSE}$ **then return** FAIL                                    ▷ **Fail**
7:    **else if** $d = \text{TRUE}$ **then return** SUCCESS                            ▷ **Success 1**
8:    **else**
9:        $i, Q \mapsto 0, \mathcal{O}_{\text{propose}}(\mathcal{P}, p)$
10:        **while** $i < \mathbf{k} \wedge |Q| > 0$ **do**
11:            $i \mapsto i + 1$
12:            $q \mapsto \text{pop}(Q)$
13:            $e \mapsto \mathcal{V}(\mathcal{P}, \{q\}, p)$                          ▷ **Propose/Backtrack**
14:            **if** $e = \text{FALSE}$ **then return** FAIL                          ▷ **Fail**
15:            **else if** $e = \text{TRUE}$ **then**
16:                $f \mapsto$ lemur_check($\mathcal{P}, q$)                           ▷ **Decide**
17:                **if** $f = \text{SUCCESS}$ **then return** SUCCESS                 ▷ **Success 1**
18:                **else if** $\mathcal{S}(\mathcal{P}, q) \wedge (\mathcal{V}(\mathcal{P}, \{\neg q\}, p) = \text{TRUE})$ **then return** SUCCESS    ▷ **Success 2**
19:                **else if** $f = \text{FAIL}$ **then** $Q \mapsto \text{join}(Q, \mathcal{O}_{\text{repair}}(\mathcal{P}, p, q, \text{FALSE}))$    ▷ **Repair 2**
20:                **else continue**
21:            **else** $Q \mapsto \text{join}(Q, \mathcal{O}_{\text{repair}}(\mathcal{P}, p, q, \text{UNKNOWN}))$    ▷ **Repair 1**
22:        **return** UNKNOWN

---

## 4    INSTANTIATING THE LEMUR CALCULUS

In this section, we present strategies for instantiating LEMUR as an automated procedure. While we showed that the LEMUR calculus is sound, there are no guarantees that it terminates. Here, we will discuss two sources of non-termination.

The first one corresponds to an unbounded sequence of suggestions for new sub-goals. Concretely, when trying to prove a particular proof goal $p$, we could get stuck if $\mathcal{V}(\mathcal{P}, \{q\}, p) = \text{UNKNOWN}$ or $\mathcal{V}(\mathcal{P}, \varnothing, q) = \text{FALSE}$ for each proposed assumption $q$. This could occur as a result of limitations in either the LLM or the verifier. One way to avoid this type of non-termination is by putting an upper bound on the number of proposed assumptions for each proof goal. That is, for any proof goal $p$, we require that $\mathcal{V}(\mathcal{P}, \{q\}, p)$ is invoked for at most **k** different values of $q$.

The second source of non-termination is the potentially unbounded depth of the trail $\mathcal{M}$. Concretely, it is possible to construct an infinite sequence of **Propose** and **Decide** transitions, where: (i) the verifier returns UNKNOWN on the current proof goal; (ii) the oracle proposes an assumption that is not invariant but implies the current proof goal; (iii) the verifier proves the implication; (iv) the assumption becomes the new proof goal; and (v) this process repeats. This can be avoided by adding a side condition to the rules requiring that properties proposed by oracles ($q = \langle \psi, l' \rangle$) must contain a smaller program line number than the one in the current proof goal ($p = \langle \phi, l \rangle$), that is,

$$\langle \psi, l' \rangle \in \mathcal{O}_*(\mathcal{P}, \langle \phi, l \rangle, \ldots) \Rightarrow l' < l \qquad \text{(Condition 1)}$$

Based on these strategies, a terminating (by Thm. 4.1 at the end of this section) and sound (by Thm. 3.1) algorithm for checking whether a property $p$ is an invariant on a program $\mathcal{P}$ is presented in Alg. 1. Alg. 1 is a recursive procedure lemur_check. It takes a program $\mathcal{P}$ and a property $p$ as inputs. If lemur_check returns SUCCESS, then the property is an invariant. If lemur_check returns FAIL, then the property is not an invariant. The function can also return UNKNOWN if the analysis is inconclusive. At a high level, Alg. 1 searches for a potential subgoal $q$ that implies the current goal $p$ (lines 9–21). If such a $q$ is identified in line 13, we recurse to prove $q$ (line 16). The while loop starting at line 10 ensures that at most $k$ attempts can be utilized to generate a new subgoal for $p$. The comments in Alg. 1 indicate which lines correspond to specific proof rules. The algorithm is sound as it only applies the rules of the calculus. A full description of Alg 1, including a proof of termination can be found in the extended version of the paper (Wu et al., 2023).

**Theorem 4.1** (Termination). *Given a program $\mathcal{P}$, and a property $p$ on the program, Alg. 1 terminates with either* SUCCESS, FAIL, *or* UNKNOWN.

Note that, Alg. 1 is one of many possible instantiations of the LEMUR calculus. One can develop alternative strategies to apply the rules, e.g., by changing the frequency of the repair rules and the **Propose** rules to balance the cost of LLM oracle calls. We evaluate one of the alternatives in the extended version of the paper (Wu et al., 2023).

## 5  EXPERIMENTS

We have presented the LEMUR calculus and described a sound and terminating algorithm based on LEMUR. In this section, we investigate the following questions:

- Can we develop a practical automated verification procedure based on Alg 1? [Yes]
- Is LEMUR competitive with existing end-to-end learning-based verification approaches? [Yes]
- Can LEMUR already prove hard benchmarks that are beyond the reach of state-of-the-art conventional program verifiers? [In several cases]

### 5.1  BUILDING AN LLM-BASED PROGRAM VERIFIER

We report on several practical considerations when building a prototype of Alg. 1. There are two types of external calls that Alg. 1 depends on. The first type is calls to $\mathcal{V}$. We use off-the-shelf verifiers in our framework that are extensively tested by the community (described in later paragraphs), so we have some expectations about their reliability and performance. On the other hand, the second type of calls, calls to LLM oracles, introduces more uncertainty, as LLMs are newer and are treated as black boxes. In our framework, the oracles $\mathcal{O}_{propose}$ and $\mathcal{O}_{repair}$ automatically prompt a GPT-family model through the OpenAI API and parse the output. We found that while GPT has great potential for generating sensible loop invariants, it still has practical limitations. We report several tactics that we found useful in practice.

- **Formatting the output**: While investigating whether chain-of-thought (CoT) (Wei et al., 2022) reasoning is useful when seeking new properties for $\mathcal{P}$ and $p$, We found that GPT's outputs, even when containing useful information, were verbose and often contained irrelevant or incorrect statements, making it difficult to extract invariants. To address this, we use in-context learning to encourage the LLM to format the output in a specific way. For example, adding `Your output should be "assert(...);// Line number"` to the prompt is sufficient for GPT to consistently generate outputs of exactly this format, without providing verbose explanations.

- **Inserting markers in the program**: We found that GPT is not good at counting program lines. Often, the generated predicate is otherwise correct, but the line number is slightly off. Unfortunately, an invariant at a wrong position is of no use to the verifier. To mitigate this challenge, we insert placeholder lines of the form `"// Line A"`, `"// Line B"` and prompt GPT to generate invariants of the form `assert(...);// Line name` (for those specific locations). As a simple practical heuristic, we insert placeholders right before loops and at the beginning of loops.

- **Ordering the proposal**: The output of an oracle call is non-deterministic for a given prompt, depending on the hyper-parameters of the call. Moreover, the oracles produce a set of properties and we need good heuristics to choose the order of trying those properties. A heuristic we found useful is to prompt GPT multiple times and order the properties by the frequency with which they are proposed (breaking ties by preferring shorter expressions). Moreover, instead of relying on string matching, we treat two proposals the same if their abstract syntax trees are equivalent.

The exact prompts are described in the extended version of the paper (Wu et al., 2023). For $\mathcal{V}$, we consider two state-of-the-art formal tools for C program verification, ESBMC (Gadelha et al., 2018) and UAUTOMIZER (Heizmann et al., 2013). The former is based on k-induction and the latter is based on predicate abstraction. ESBMC and UAUTOMIZER were the top two performing non-portfolio solvers in the reachability track of the most recent edition of the software verification competition (SV-COMP (Beyer, 2023)). Furthermore, UAUTOMIZER was the overall winner. By default, we impose a 30-second time limit for each invocation of the verifier. That is, if the verifier does not terminate within 30 seconds, then the verifier returns UNKNOWN.[3]

---

[3]The source code and the benchmarks are publicly available at https://github.com/wu-haoze/Lemur-program-verification.

| Configurations | Solved | Time (s) | # proposal |
|:---:|:---:|:---:|:---:|
| Code2Inv | 92 | – | - |
| ESBMC | 68 | 0.34 | 0 |
| LEMUR(GPT3) | 103 | 35.6 | 8.6 |
| LEMUR(GPT4) | 107 | 24.9 | 4.7 |

(a) The Code2Inv benchmarks.

| Configurations | Solved | Time (s) | # proposals |
|:---:|:---:|:---:|:---:|
| UAUTOMIZER | 1 | 824.3 | 0 |
| ESBMC | 1 | 675.7 | 0 |
| LEMUR(GPT3) | 14 | 162.2 | 8.5 |
| LEMUR(GPT4) | 25 | 234.5 | 7.2 |

(b) The 47 SV-COMP benchmarks.

Table 1: Solved instances by ESBMC, LEMUR, and Code2Inv (1a) or UAUTOMIZER (1b) on two benchmark sets. We also report the average time and number of proposals for solved instances.

## 5.2 LOOP INVARIANT GENERATION BENCHMARKS

A prominent approach in learning-based end-to-end program verification is Code2Inv (Si et al., 2020), which uses reinforcement learning to train an invariant synthesizer to propose loop invariants. At a high level, to infer a loop invariant for a given program, Code2Inv learns a generative neural network whose goal is to repeatedly propose candidate invariants until an automated reasoning tool determines that the candidate is correct. Throughout this process, the generator is fine-tuned using the feedback from the automated reasoning tool. In this section, we study how LEMUR compares with this approach on the same benchmark set considered in the original Code2Inv work. The benchmark set contains 133 benchmarks, each containing a C program and a property expressed as an assert statement in the program. Each program contains a single loop and each loop can have nested if-then-else blocks (without nested loops). The assertion to check is always after the loop. Code2Inv is designed to generate invariants at the beginning of the while loop. We prompt the oracles to generate invariants at the same location.

We use the k-induction-based verifier, ESBMC, to check the implication (line 13 in Alg. 1) which aligns with the verification procedure used in Code2Inv. We report the number of solved instances as well as the number of failed suggestions (either the suggestion itself cannot be verified or ESBMC times out on the implication check). As a point of comparison, we report the statistics from the original Code2Inv approach. Code2Inv was given a one-hour timeout. In addition, we also report ESBMC's performance on this set of benchmarks. The result is shown in Table 1a (the Time column shows the average time in seconds to solve a benchmark).

With a 10-minute timeout, ESBMC alone can solve 68 problems. On the other hand, LEMUR(GPT4) can solve 107 problems within the same time limit. Surprisingly, this approach solves more instances than Code2Inv, which is specifically designed for invariant synthesis tasks. For problems unsolved by ESBMC but solved by LEMUR(GPT4), a histogram of the values of $Log_2$ of the number of proposals is shown in Fig. 3. While in most cases, Alg. 1 can produce the correct proposals within 4 attempts, there are benchmarks for which LEMUR(GPT4) requires many iterations to find the desired loop invariant, e.g., one of the benchmarks took 177 proposals. In addition, we experimented with the GPT-3.5 turbo LLM model, denoted LEMUR(GPT3), as shown in Table 1a. Note that LEMUR(GPT3) solved four fewer benchmarks and required more time and more calls to the GPT-3.5 turbo oracle.

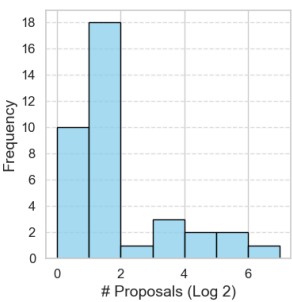

Figure 3: Nb. of proposals for LEMUR(GPT4) to solve a Code2Inv benchmark.

## 5.3 SOLVING HARD SV-COMP BENCHMARKS

Next, we study LEMUR's ability to solve hard benchmarks from the 2003 edition of SV-COMP (Beyer, 2023). We focus on benchmarks with less than 150 tokens (after removing comments and unnecessary headers, and after applying clang-formatting).We select 47 benchmarks that ESBMC and UAUTOMIZER are unable to solve within 10 minutes. The property is expected to hold in all benchmarks. We use a 15-minute per-instance timeout.

The results are shown in Table 1b. Impressively, with the guidance of the proof goals suggested by the LLM, LEMUR(GPT4) is able to solve 25 of the 47 SV-COMP benchmarks. While ESBMC and UAUTOMIZER can each solve only 1 benchmark. Upon closer examination, 8 of the solved

instances contain two loops and 5 contain three or more loops. This suggests that LEMUR is capable of (i) handling programs with more than one loop; and (ii) boosting the performance of state-of-the-art conventional C program verifiers.

The average number of proposals before solving a problem is higher compared to the Code2Inv benchmarks (7.2 vs. 4.7). Fig. 4 sheds more light on the behavior of LEMUR(GPT4). We observe that 12 of the 26 solved instances require at least 5 proposals in total.

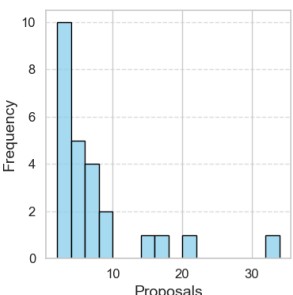

Figure 4: Nb. of proposals for LEMUR(GPT4) to solve an SV-COMP benchmark.

We found that LLM oracles can produce surprisingly insightful loop invariants which are difficult for conventional formal methods to synthesize. While predicate-abstraction-based techniques typically generate predicates that involve only the operators and values in the program and follow a particular template, LLMs are not constrained by these limitations. For example, for the program in Fig. 2, GPT-4 can consistently generate `x%4==0` as the loop invariant although the modulo operator is not present in the program. In another example, the LLM correctly understands the range of an `unsigned char` and suggests variable bounds as the assumption, which ends up being the key to proving the property. This example is shown in the extended version of the paper (Wu et al., 2023). There are also several cases where the LLM generates disjunctive invariants that precisely characterize the behavior of the loops.

Finally, we observe that LEMUR(GPT4) significantly outperforms LEMUR(GPT3) across all metrics. This suggests that the choice of oracle is also crucial for performance. Additional experiments are presented in the extended version of the paper (Wu et al., 2023). They include running the baseline solvers with a 12 hour timeout, using a configuration in which repair rules are not employed in Alg. 1, and running LEMUR multiple times to account for the stochasticity of the oracles.

## 6 DISCUSSION OF LIMITATIONS AND EXTENSIONS

In this work, we propose a novel framework, LEMUR, which combines automated reasoning and LLMs. To the best of our knowledge, LEMUR is the first framework to provide a theoretical foundation for such an integration via a formal calculus. We also implemented LEMUR as a fully automated framework and demonstrated its efficiency on standard benchmark sets. We conclude by discussing the current limitations of LEMUR, which also point to future research directions.

As mentioned above, the practical performance of LEMUR depends on two types of external calls: the verifiers and the LLMs. Any improvements in these tools should translate into LEMUR improvements. Currently, modern verifiers are capable of handling only relatively small programs (see SV-COMP'23 (Beyer, 2023)). Interestingly, even when provided with a strong invariant, they sometimes cannot solve the verification problem. One research direction that we envision is to customize LEMUR to a particular back-end verifier to obtain better performance and solve larger programs.

We also note that LEMUR primarily focuses on imperative languages. Extending it to functional languages is a direction for future research.

While our experience with LLMs was largely positive (see Section 5.1 for a discussion of limitations that have already been at least partially addressed), there are more interesting challenges to tackle. First, current LLMs have token limits, and many practical programs exceed those limits. Second, it is sometimes challenging for LLMs to generate complex logical formulas such as those with nested if-then-else expressions. We believe that to overcome this limitation, we need to (i) develop a prompting language for LLM invariant generation, and (ii) fine-tune LLMs for invariant generation tasks using this language. Third, reasoning about programs with multiple loops remains challenging for LLMs. We believe fine-tuning could help address this challenge. Fourth, we observed that the performance of LEMUR may vary depending on the LLM oracle. For example, our experiments demonstrate that GPT-4 is superior to GPT-3.5 turbo on tested benchmarks. Finally, due to the limitations of LLMs and automated reasoners, our framework does not yet offer a significant boost for complex properties of real-world C libraries. However, a modular approach, where large parts of the program are abstracted and summarized in the form of pre- and post-conditions, can benefit from frameworks like LEMUR.

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
