# OpenReview forum: "Lemur: Integrating Large Language Models in Automated Program Verification"
_ICLR.cc/2024/Conference — ICLR 2024 poster_

### Official Review · Reviewer_mzY5 · 2023-10-31

**Soundness:** 4 excellent
**Presentation:** 4 excellent
**Contribution:** 4 excellent
**Rating:** 8
**Confidence:** 5

**Summary:**

The paper presents a principled approach for combining LLMs with automated reasoning tools to perform automated program verification. Automated program verification typically proceeds by breaking down the overall proof goal into simpler proof sub-goals---each sub-goal establishes some invariant of the program which can help in proving/disproving the overall program assertion. The proposed technique uses LLMs to suggest candidate invariants as proof sub-goals; whether the suggested invariants hold or not is posed as a query to existing automated program reasoning tools. While the general paradigm of guessing and then checking invariants is well-known in the program verification literature, the paper presents a proof calculus that cleanly formalizes this style of reasoning and develops an algorithm based on this calculus. The calculus is proven sound and the algorithm is shown to be terminating under suitable conditions. Most importantly, the presented algorithm outperforms state-of-the-art automated program verification tools on standard benchmarks.

**Strengths:**

1. This is a very well-written paper and presents a nice, clean formalization of the guess-and-check style of program reasoning via the Lemur calculus.

2. The algorithm that operationalizes the calculus is elegant and easy to understand.

3. Use of LLMs to suggest candidate program invariants is an obvious idea but it has been very well manifested into practice, both mathematically and algorithmically,  by this work.

4. Some of the prompting tricks used to get the invariants from the LLM, such as placeholder lines, are interesting in their own right.

5. Most importantly, the empirical results are very promising and suggest that LLMs and verification tools can be fruitfully combined.

**Weaknesses:**

1. This is a relatively minor comment but the formalization and the algorithm seem geared towards imperative languages where the notion of associating a property/invariant with a line number makes natural sense. It would be helpful to acknowledge that the proposed calculus might not necessarily be applicable to all programming languages.

2. There has a large body of literature on data-driven techniques for learning loop invariants [1,2]. Unlike Code2Inv and Lemur, these past works use dynamic values of loop variables. Adding references to this body of work would help paint a fuller picture about this area.

[1] Garg, P., Neider, D., Madhusudan, P., & Roth, D. (2016). Learning invariants using decision trees and implication counterexamples. ACM Sigplan Notices, 51(1), 499-512.

[2] Sharma, R., Gupta, S., Hariharan, B., Aiken, A., Liang, P., & Nori, A. V. (2013). A data driven approach for algebraic loop invariants. In Programming Languages and Systems: 22nd European Symposium on Programming, ESOP 2013, Held as Part of the European Joint Conferences on Theory and Practice of Software, ETAPS 2013, Rome, Italy, March 16-24, 2013. Proceedings 22 (pp. 574-592). Springer Berlin Heidelberg.

**Questions:**

1. For the benchmark of difficult programs from SV-COMP, where are the placeholder lines added? Is it after every line in the program? If not, doesn't the location of the placeholder leak information, potentially aiding Lemur?

2. It seems like the text and the formal expression for the second bullet point of Proposition 2.1 do not line up. Shouldn't the implication in the formal expression go the other way?

3. I don't think the last line on Page 2 is correct. A program without loops can have an unstable property because the program can still have multiple paths due to branches. Also, how can a property that is not an invariant be stable? Is this the case where stability is due to the property always being false?

4. Is the time in Table 1 in seconds? When you say timeout of 1 hr or 10 minutes, is that the timeout for the entire dataset or each program?

5. Does each bar in Figure 4 correspond to a single value for the number of proposals or a set of values? It seems like it is the latter case. It would help to clarify the figure.

---

> ### Author Response · Authors · 2023-11-20
>
> Thank you for your comments and suggestions!
>
>
> **Q: This is a relatively minor comment but the formalization and the algorithm seem geared towards imperative languages where the notion of associating a property/invariant with a line number makes natural sense. It would be helpful to acknowledge that the proposed calculus might not necessarily be applicable to all programming languages.**
>
> We will clarify that our approach is designed for imperative languages. Thank you for the clarification suggestion!
>
>
> **Q: There has a large body of literature on data-driven techniques for learning loop invariants [1,2]. Unlike Code2Inv and Lemur, these past works use dynamic values of loop variables. Adding references to this body of work would help paint a fuller picture about this area.**
>
> We will include references to these papers and discuss them in the 'Related Work' section. Thank you!
>
>
> **Q: For the benchmark of difficult programs from SV-COMP, where are the placeholder lines added? Is it after every line in the program? If not, doesn't the location of the placeholder leak information, potentially aiding Lemur?**
>
> We automatically insert placeholders at the beginning of each loop and right before the loop, which we deem a generally good heuristic in practice. We do not think we are leaking any information specific to the benchmarks to Lemur.
>
>
> **Q: It seems like the text and the formal expression for the second bullet point of Proposition 2.1 do not line up. Shouldn't the implication in the formal expression go the other way?**
>
>
> Thank you for spotting that! Yes, it should be the other way in the formal expression.  We updated the paper.
>
>
>
> **Q: I don't think the last line on Page 2 is correct. A program without loops can have an unstable property because the program can still have multiple paths due to branches.**
>
> We say the property is stable, if within a single execution of the program, the property always evaluates to true or always evaluates to false. If there are no loops, the property will be evaluated at most once. This also holds when the program has branches, as only one path will be executed.
>
>
> **Q: Also, how can a property that is not an invariant be stable? Is this the case where stability is due to the property always being false?**
>
> The property is stable as long as it evaluates to the same truth value within a particular execution (please see the clarification on the definition of stability above). Therefore, if the property is always false, then the property is stable. However, it is also possible for a property to be stable if it holds in some executions, but does not hold in some other executions.
>
> Consider an illustrative example:
>
> ```
> Line 1: x=rand(0,10)
> Line 2: assert(x==1)
> ```
>
> The property in Line 2 is stable. This property only holds for some executions.  We will add an example to the paper to illustrate the stability definition.
>
> **Q: Is the time in Table 1 in seconds? When you say timeout of 1 hr or 10 minutes, is that the timeout for the entire dataset or each program?**
>
> Yes, the time is in seconds.  And the timeout is per program.
> We will clarify that in the paper.
>
> **Q: Does each bar in Figure 4 correspond to a single value for the number of proposals or a set of values? It seems like it is the latter case. It would help to clarify the figure.**
>
>
> Each bar in Figure 4 corresponds to a set of values.  We will clarify this in the text.

---

> > ### Comment · Reviewer_mzY5 · 2023-11-21
> >
> > Thank you for the response and the clarifications, specially, about the stability property.

---

### Official Review · Reviewer_ERFy · 2023-10-31

**Soundness:** 2 fair
**Presentation:** 3 good
**Contribution:** 2 fair
**Rating:** 5
**Confidence:** 3

**Summary:**

This paper proposes an approach to reachability verification, dubbed Lemur, which combines the invariance generation capabilities of LLMs with standard verification tools. The resulting procedure, which is sound albeit incomplete, comes with theoretical guarantees.
Lemur equipped with GPT4 can solve benchmarks that are currently out of reach of state-of-the-art standard reachability verification tools.

**Strengths:**

- Clear presentation
- Reachability is a pivotal problem in program verification
- Promising preliminary results
- (To the best of my understanding) Lemur's calculus and template algorithm can accomodate different combinations of invariant generators and verifiers

**Weaknesses:**

My main concerns are related to the empirical evaluation of Lemur.
The combination of LLMs and standard verification procedures is indeed promising, but I am left wondering how much the results are reliant on the specific LLM GPT4.

This is particularly important considering the economic cost of reproducing the experiment, which can be prohibitive for some.

If I understand correctly, Lemur is a template algorithm. I would then expect to see results for LEMUR(X, Y), where X = {GPT4, other free LLMs, other invariant generation techniques (Code2INV maybe?), and Y = {UAUTOMIZER, ESBMC}.

**Questions:**

- How much is Lemur performance reliant on the invariant generator quality?
- How does Lemur fare without GPT4 as an invariant generator?
- Can Lemur be used with non-LLM invariant generators such as Code2Inv?

--------
Minors:

The derivation rules in Fig.1 are not 100% clear to me. What is the semantic of operator ": :"?

Figures 3 and 4 should have different (and self-explanatory) captions, i.e. explicitly mention the referred benchmarks.

---

> ### Author Response · Authors · 2023-11-20
>
> Thank you for your comments and suggestions!
>
>
> **Q: How much is Lemur performance reliant on the invariant generator quality? How does Lemur fare without GPT4 as an invariant generator?**
>
> We agree that this is an interesting question of how sensitive Lemur is to an LLM choice. We made a decision to build the first version of Lemur on GPT4 which are arguably the most powerful models and absorb the cost of using these models.
>
> *We ran additional experiments with the GPT-3.5 turbo model and updated the experimental evaluation in Tables 2a and 2b. We observe that GPT-4 outperforms GPT-3.5 turbo, especially on SV-COMP benchmarks. Based on these results, we hypothesize that the performance might drop if we use other open-source models (but this can change as these models are constantly improving).*
>
> **Q: This is particularly important considering the economic cost of reproducing the experiment, which can be prohibitive for some.**
>
> We agree with the reviewer that it might be costly to use. To be more concrete, we spent ~2 cents per call for GPT4. We chose to use the best LLM model out there to see what is the potential of such integration. Our total budget was less than $1000 for the project.
>
> **Q: Can Lemur be used with non-LLM invariant generators such as Code2Inv?**
>
> Code2Inv is trained to find an invariant at a fixed location in a program, which is not suitable for our framework.
> Our framework requires an LLM oracle to be able to generate an invariant at any point in the program.
>
> In principle, we can use any LLM and verification tools. For example, we added new results for GPT3.5 oracle instead GPT4 that are presented in the paper. As we mentioned above, our new results show that GPT-4 outperforms GPT-3.5 turbo on the tested benchmarks.
>
> We are happy to run more experiments with open-source LLMs in the final version of the paper but we will not be able to provide these results during rebuttal, unfortunately.
>
>
>
> **Minors Q: The derivation rules in Fig.1 are not 100\% clear to me. What is the semantic of operator ": :"?**
>
> :: means that a list M is a concatenation of sublists, e.g. M = sublist1::sublist2
>
> For example, we often use the following notation in Figure 1.
> ```
> M= M'::p
> ```
> This means that a trail M is a concatenation of a trail M' and p, where p is the last element of M. We will further clarify this in the paper.
>
>
> **Minors Q: Figures 3 and 4 should have different (and self-explanatory) captions, i.e. explicitly mention the referred benchmarks.**
>
> Thanks, we will fix.

---

> > ### Comment · Reviewer_ERFy · 2023-11-22
> > **Response to the authors**
> >
> > Thank you for your reply. I think that the extra experiment with a different LLM are valuable.
> >
> > I tend to agree with Reviewer 2ahy's concerns on the fairness of a runtime comparison between the baselines and LEMUR using a LLM oracle running on external hardware. I understand that running the LLM oracle locally is not completely solving the issue, since most reachability tools do not make use of the GPU anyway (to the best of my understanding). Yet, in my opinion having LEMUR running completely on local would be more significant for practitioners, giving some insight on the performance gains that we can expect on limited hardware resources.

---

> > > ### Author Response · Authors · 2023-11-22
> > >
> > > **Q: Running Lemur locally/free**
> > >
> > > To run Lemur locally, we need to use an open-source model. As our experiments show, even using GPT-3.5 instead of GPT4 degrades the benefits of synergy between an LLM and verification. We do not believe we can obtain better results with open-source models.
> > >
> > > We note that the cost of calling GPT is not high, only 2 cents per call. For safety-critical applications, such cost is easily justifiable given the cost of the potential bugs.

---

### Official Review · Reviewer_YmWA · 2023-11-01

**Soundness:** 2 fair
**Presentation:** 1 poor
**Contribution:** 2 fair
**Rating:** 5
**Confidence:** 3

**Summary:**

The paper describes a system for program verification that queries CHATGp4 for axioms.
The idea is interesting and the results excellent,
The main problem of the paper is that it is written in a way that is much more suitable fior a CAV venue  than fior ICLR. The authors carefully and fomally describe the program  analysis framework, but they take a loose approach about GPt

**Strengths:**

The main strength of the paper are the results in Section 5\

.

**Weaknesses:**

Presentation

Only good results are shown? How likely is the fail case? ChatGPT seems a magic box, Above al, the way you present your work makes it unnecessarily hard to understand ypur ideas.

**Questions:**

A small point: there is another lemur system in LLMs

"An important research question is whether modern AI models are capable of understanding ." .. indeed, but can you claim you do that?

Are you the first to ask LLMs to propose sub-goals?

I'd much prefer a better survey that would make your contributions clear, and a graphical descriotion of the system that would allow me to position the different componrrntd

I understand the process is fully automated?

 "  -Finally, if the verifier V proves that the original property is not an invariant" -> is the verifier cmplete?

Arguabythe prompts are what makes everything else worthwhile. Yet, they are in appendix?

---

> ### Author Response · Authors · 2023-11-20
>
> Thank you for your comments!
>
>
> **Q: "An important research question is whether modern AI models are capable of understanding ." .. indeed, but can you claim you do that?**
>
> It has been shown in [1] that LLMs have potential understanding programs. We take these ideas to a new level, showing that we can leverage this understanding to help state-of-the-art verification tools verify more C programs. So, we believe that we make a step toward showing AI understanding of realistic C programs.
>
> [1] Kexin Pei, David Bieber, Kensen Shi, Charles Sutton, and Pengcheng Yin. Can large language models reason about program invariants? ICML23
>
>
> **Q: Are you the first to ask LLMs to propose sub-goals?**
>
> Yes, we are the first to propose this idea.  We outlined our contributions in Introduction, we will emphasize that we are the first to propose these techniques.
>
> **I'd much prefer a better survey that would make your contributions clear, and a graphical descriotion of the system that would allow me to position the different componrrntd**
>
> We will include a table that compares learning-based augmented verification techniques in Appendix E.
>
>
> **Q: I understand the process is fully automated?**
>
> Yes, that is correct. The process is fully automated. We will highlight this point in our contributions.
>
>
> **Q: " -Finally, if the verifier V proves that the original property is not an invariant" -> is the verifier complete?**
>
> As discussed in the paper, V needs to be sound and we do not assume it is complete. For example, if the program deals only with bit-vectors, then the underlying procedure is complete. But if the program contains arrays, then the verification problem is undecidable and the verifier is incomplete.
>
>
>
> **>? Arguabythe prompts are what makes everything else worthwhile. Yet, they are in appendix?**
>
>  For our framework, we have a small set of predefined template prompts that correspond to our rules, which is why they are included in the appendix. We kindly disagree with the reviewer that the prompts are key components in our framework. While we did put effort into designing these templates, the main component is the theoretically sound integration of LLMs with automated reasoners. If space permits, we are happy to move the prompt templates to the main text.

---

### Official Review · Reviewer_2ahy · 2023-11-06

**Soundness:** 4 excellent
**Presentation:** 3 good
**Contribution:** 2 fair
**Rating:** 5
**Confidence:** 5

**Summary:**

This paper presents an interesting way to use LLMs and automated program verification tools synergistically to prove properties of programs that may otherwise be difficult to prove by the verification tools themselves.  The core idea consists of using an LLM to generate potential assumptions that may help an automated program verifier discharge the proof goal.  Once such assumptions are found, the effort of the program verification tool is directed to proving the assumptions themselves.  The authors present a set of sound rules to transform "configurations" of the proof process involving LLM calls and program verifier calls.  The authors also present some heuristics for LLM prompt generation, and for deciding how to prioritize multiples responses that may be provided by the LLM.  Finally, the authors present experimental results that show the promise of the proposed technique vis-a-vis state of the art program verification tools.

**Strengths:**

The paper is clearly written, modulo some typos.  This helps in taking the reader along with the flow of the presentation.  The core ideas are illustrated with a running example -- this helped me follow the ideas without much difficulty. The idea of using an LLM to propose assumptions that can then be used to simplify a verification task is promising and the experiments demonstrate this.  Being able to out-perform the winning entries in SV-COMP is a significant achievement, and I believe this sufficiently demonstrates the promise of the approach.

**Weaknesses:**

The set of rules formulated by the authors doesn't add much value to the paper.  Algorithm 1 could itself have been presented directly, with explanations for the different steps.  In my opinion, the rules are kind of a force-fit to the paper.
It is good that the authors try to show that the proof rules are sound; however the soundness proofs are simple and not very insightful.  Indeed, finding a sound set of rules for simplifying/decomposing program verification is often not the difficult part of the process.  The more technically challenging part is to have completeness results for some subclass of programs/properties.  Unfortunately, the authors don't even try to do this.
Since there are practical limits to prompt lengths for an LLM like GPT or GPT-4, this sets limits to how large a program can be verified using the proposed technique.  This is highly undesirable.  It would have been better if the authors attempted decomposition of the problem (perhaps guided by an LLM) and then applied the proposed technique to each decomposed part, and then stitched the results back together.  Unfortunately, the authors completely avoid any such attempt/discussion.
SV-COMP has several tracks for verification.  The total count of programs in SV-COMP is significantly larger than that reported by the authors. It is not clear whether some cherry-picking was done in selecting the examples.
An LLM like GPT-4 makes use of immense computational power at the server end to come up with proposals/suggestions quickly.  It is not fair to discount this computational effort completely when comparing with the performance of other solvers that do not make LLM calls.  As a consequence, I believe the time comparison is not fair.
A method like the one proposed, without any considerations of how the LLM is trained, may give dramatically different results based on what LLM is being used.  Therefore, a discussion on training the LLM should have been included.

**Questions:**

1.  What actual value does the set of rules provide to this work?  Wouldn't it have sufficed to have Algorithm 1 directly, along with an explanation of the steps?
2.  SV-COMP contains many, many more benchmarks.  How does LEMUR compare with UAutomizer or other tools on the other benchmarks?  Why are so few benchmarks chosen for comparison?
3.  For what kinds of <program, property> combinations do LLMs fare badly as far as suggesting assumptions is concerned?  In some sense, this ought to depend on what kinds of data it has been trained on.   I didn't see any discussion in this regard -- I would tend to think that there are <program, property> combinations for which LLMs will have a very difficult time generating good assumptions.
4. An LLM like GPT-4 makes use of immense computational power at the server end to come up with proposals/suggestions fast.  How do you propose to factor this in your comparisons for an apples-to-apples comparison.  Neither ESBMC not UAutomizer were provided access to such computational power; so how is the comparison fair?
5. What are the assumptions on the training of the LLM that are being used?  Will the proposed method work with one's privately trained LLM?

**Details Of Ethics Concerns:**

No ethics concerns

---

> ### Author Response · Authors · 2023-11-20
>
> Thank you for your comments and suggestions!
>
>
> **Q: What actual value does the set of rules provide to this work? Wouldn't it have sufficed to have Algorithm 1 directly, along with an explanation of the steps?**
>
> Our derivation rules define an underlying proof calculus, while Algorithm 1 represents one of many possible strategies for applying these rules. We believe that the proof system is significant, as it formally defines the interaction between the verifier and oracles in a sound and modular way. Note that we designed the rules to have a *clear separation of responsibility between two powerful technologies* to take the best of both worlds. We also have introduced the notion of *stability* and a rule that leverages this new concept. We believe that we are the first to propose such formalization, which can be built upon in the future.
>
>
> **Q: Scalability of verification**
>
>
> As we mentioned in the "Discussion of Limitations" section, our approach currently works with smaller programs (<= 150 tokens after removing comments, unnecessary headers, and clang-formatting), and a decomposition approach might be good for scaling to larger programs, as the reviewer also suggested.
>
> Decomposing a large program into smaller pieces for verification purposes is known to be challenging. Using LLM to perform this task requires the representation of the program in some compact form, a specification language for pre-/post- conditions, and an efficient verification procedure to validate the correctness of those conditions.
>
>
> **Q: SV-COMP contains many, many more benchmarks. How does LEMUR compare with UAutomizer or other tools on the other benchmarks? Why are so few benchmarks chosen for comparison?**
>
> We call UAutomizer/ESBMC first, before augmenting a verifier with GPT-based generated invariants. Hence, in theory, we can solve at least as many instances as UAutomizer can, provided there is an appropriate timeout for the initial call (Algorithm 1, line 5).
>
> However, our main goal was to demonstrate that GPT can aid UAutomizer/ESBMC in solving problems that *neither of these solvers can solve independently*. We selected 50 smaller benchmarks from reachability track that these tools cannot solve and demonstrated that GPT aid can help with 26 of these challenging benchmarks. We find it impressive that LLMs help to solve these benchmarks, even if we could only solve about half of them.
>
>
> **Q:  For what kinds of <program, property> combinations do LLMs fare badly as far as suggesting assumptions is concerned? In some sense, this ought to depend on what kinds of data it has been trained on. I didn't see any discussion in this regard -- I would tend to think that there are <program, property> combinations for which LLMs will have a very difficult time generating good assumptions.**
>
> As shown in the experiment, there were 24 out of the 50 challenging benchmarks that we could not solve. Upon closer examination, we observed that it is challenging to 1)  generate complex invariants involving if-then-else logic; 2) generate invariants where a program contains multiple loops. We will expand the discussion of limitations section in the paper to clarify this point.
>
>
> **Q:  An LLM like GPT-4 makes use of immense computational power at the server end to come up with proposals/suggestions fast. How do you propose to factor this in your comparisons for an apples-to-apples comparison. Neither ESBMC not UAutomizer were provided access to such computational power; so how is the comparison fair?**
>
> *We do not see our work as a competitor to modern program verifiers. Our goal is to explore whether LLM can help these tools perform better.*
>
> *Moreover, we point out that most existing verifiers (e.g., ESBMC and UAutomizer) are not designed to leverage immense computational power like GPUs or cloud computing, and our approach provides such means to empower the verifiers.*
>
>
> **Q:  What are the assumptions on the training of the LLM that are being used? Will the proposed method work with one's privately trained LLM?**
>
> We used OpenAI GPT-4 API service (we will also add new results for the OpenAI GPT-3.5 turbo).  Unfortunately, exact training data is not known for GPT4. It might be the case that SV-COMP benchmarks were part of the training set but invariants are not available publically as far as we know.

---

> > ### Comment · Reviewer_2ahy · 2023-11-22
> > **Thanks for the response**
> >
> > Thanks to the authors for responding to my questions. Unfortunately, some crucial questions (for me) remain unsatisfactorily answered.
> >
> > The answer to the question about the actual value of set of rules is unsatisfactory, in my opinion.  Proving soundness of the algorithm wouldn't have been hard even without the rules.  The real value of the system of rules would have been to help prove some kind of relative completeness result.  Unfortunately, the authors don't do this.
> >
> > The authors mention that they do not see their work as a competitor to modern program verifiers.  And yet, they compare the number of cases solved by their tool (which uses a GPT with unspecified computing power) with state-of-the-art verifiers *within a given limit of computational power* (all SV-COMP participants are given only limited computational power).  I don't see how this is an apples-to-apples comparison. Can the authors really assert that UAutomizer or ESBMC would not be able to solve the problems on which the authors demonstrate their strengths, if UAutomizer/ESBMC had access to additional computing resources?  Sans this, how do we really know whether use of LLMs is the best way to go about spending additional compute power for verification.  The $1000 that the authors spent on GPT-4 would have given enough computing power on AWS or Google Cloud, for UAutomizer/ESBMC to crank for sufficiently longer (and perhaps solve all the problems).  I really don't see why we must try to force fit LLMs into verifiers if they don't present a cost-effective solution.

---

> ### Author Response · Authors · 2023-11-23
>
> Thank you for your comments!
>
> **Q: Completeness of calculus rules**
>
> *We strongly disagree that the value of abstract calculus is only the proof of completeness*.
>
> Many highly impactful abstract calculus in SMT (e.g., notably Model-Constructing Satisfiability Calculus [1])  do not attempt to establish completeness results. In the context of formal methods, abstract calculus has the following benefits:
> * A separation between the implementation and the theory,
> * Highlighting the key features of the procedure in a modular manner,
> * Providing formalized grounds for future extension.
>
> [1] A Model-Constructing Satisfiability Calculus,  Leonardo de Moura & Dejan Jovanović
>  https://link.springer.com/chapter/10.1007/978-3-642-35873-9_1
>
>
> Would the review agree that our calculus serves these purposes?
>
>
> **Q: Running program verification performance on a cloud/more resources**
>
> We are not aware of any reported significant performance gains in software verification tasks with parallelization. In fact, a very recent paper shows that achieving parallelization of SMT solvers — the underlying solvers for both UAutomizer and ESBMC — is not trivial. For example, using 64 cores resulted in less than 2 times speed-up [2].  We therefore do not expect that allocating more computational resources to the verifier will significantly improve the number of instances solved within the given time bound.
>
> [2] Partitioning Strategies for Distributed SMT Solving
> Amalee Wilson, Andres Noetzli, Andrew Reynolds, Byron Cook, Cesare Tinelli, and Clark Barrett,
> FMCAD'23

---

> > ### Comment · Reviewer_2ahy · 2023-11-23
> > **Response to authors**
> >
> > I thank the authors for their response.
> >
> > I believe my comment about abstract rules has been mis-interpreted.  Let me clarify.  I am well aware of the value of formalization of abstract rules for various algorithms in the area of formal methods.  It would be unfair to interpret my comment as saying that the only value of abstract rules *in any context* is in proving completeness.  My comment was in the specific context of Lemur.  The algorithm presented by the authors can be proved sound without reference to the rules, in my opinion.  So, proving soundness of the algorithm is not a strong value addition (if at all) of the rules.  I agree that abstraction of steps allows scope for further generalizations in an elegant way.  A sterling example of this is [1], where Simplex was generalized to Reluplex.   The paper on model-constructing satisfiability calculus for SMT theories that the authors point out achieves a whole bunch of objectives through the abstract rules: it presents a unifying framework for a large collection of model-based SMT decision procedures, allows DPLL(T) and the nlsat proof calculus to be combined, and also allows a simpler correctness proof than nlsat.  I don't see any such value additions of the Lemur proof calculus beyond the correctness proof of the Lemur algorithm, and even there I don't see any simplifications that are obtained by the rule abstraction.  I agree that the rules in the case of Lemur provide modular highlighting of features of the algorithm.  However, I disagree that in the specific case of Lemur, the rules provide a significant separation between theory and implementation, beyond what the steps of the algorithm already provide.  This is why I had commented that a potential value addition of the set of rules *in the specific case of Lemur* could have been a step towards proving completeness.  To interpret this as saying that the only value of rule formalization *in all cases* is to prove completeness is completely taking my comment out of context.
> >
> > [1] "Reluplex: An Efficient SMT Solver for Verifying Deep Neural Networks",  Guy Katz, Clark Barrett, David L. Dill, Kyle Julian & Mykel J. Kochenderfer, CAV 2017
> >
> > Regarding the comment about use of parallelization, I wasn't referring only to parallelization.  If we are willing to spend more resources on computing, we might as well use the additional resources to run a tool for longer duration or with more memory than the limits imposed by SV-COMP.  Are the authors saying that this doesn't help in improving the tally of verification results?  This reviewer has to respectfully disagree, based on first-hand experience.  In fact, the paper pointed to by the authors already shows improvement, though not as much as a nicely parallelizable task would show.  This just shows that parallelization is difficult, though it does provide improvements.  Even if we go by the parallelization results pointed to by the authors, wouldn't a better comparison be to see how ESBMC or UAutomizer fares with the same $ budget used for GPT calls (as in Lemur) and for additional cores on a cloud platform? Unless we take the cost-benefit analysis of this into account, it may be a bit premature to assert that Lemur helps improve existing tools.
> >
> > Having said all of the above, I agree that the paper presents some interesting research directions.  I hope the authors take my comments in a constructive manner.  I believe this work would be much stronger (and would address some important research questions) if some of the concerns raised above are attended to, and if a fairer comparison taking into account the cost of LLM calls is presented.

---

> > > ### Author Response · Authors · 2023-11-23
> > > **Algorithm 1/derivation rules**
> > >
> > > We thank the reviewer for their clarifications. We now understand better what is the reviewer's concern regarding the value of the rules in proving soundness of Algorithm 1.
> > >
> > > We agree with the reviewer that one can directly prove the soundness of Algorithm 1. However, if someone proposes an alternative algorithm based on our rules, they have to re-prove soundness.
> > >
> > > We chose to prove the soundness of the calculus instead of just Algorithm 1 because this allows us to cover all possible proof derivations, so that any alternative algorithm based on our rules is sound by construction.

---

> > > ### Author Response · Authors · 2023-11-23
> > > **Computational resources**
> > >
> > > We thank the reviewer for the clarifications.
> > >
> > > Regarding *longer timeout*, we have run both ESBMC and UAutomizer on the benchmarks with a 1-hour timeout, and no additional instances were solved. While we are happy to report results with a longer timeout in the final version, we believe it will unlikely help in most of these benchmarks, as we observed that the verifiers typically got stuck in the call to the underlying SMT solver.
> > >
> > > Regarding *parallelization*, we agree that the comparison that the reviewer suggested would be an interesting experiment. However, we are not aware of cloud/massively parallel adaptations of ESBMC or UAutomizer that are readily available. Creating massively parallel automated reasoners is a challenging research problem on its own and we believe it is out-of-scope of our work.

---

### Author Response · Authors · 2023-11-21
**Summary of revisions**

We thank all reviewers for their helpful feedback, comments, and suggestions!

We addressed each review's concerns individually and uploaded a revised version of the paper that incorporates the reviewers' suggestions (all changes have been highlighted for clarity).

Moreover, we conducted additional experiments with GPT-3.5 turbo. Below is a snapshot of these results, providing insights into the comparison between the GPT-4 (denoted LEMUR (GPT-4)) and GPT-3.5 turbo (denoted LEMUR (GPT-3)) oracles.

Results on Code2Inv benchmarks:

|Configurations  |Solved | Time (s) |  # proposals|
|---------------------|----------|------------|------------------|
|LEMUR (GPT-4):| 107| 24.9| 4.7|
|LEMUR (GPT-3):| 103| 35.6| 8.6|

Results on SV-COMP benchmarks:

|Configurations  |Solved | Time (s) |  # proposals|
|---------------------|----------|------------|------------------|
   | LEMUR (GPT-4): | 26| 140.7| 9.1|
   | LEMUR (GPT-3):| 15| 223.7| 12.8|


Overall,  LEMUR (GPT-4) outperforms  LEMUR (GPT-3) on both benchmark sets.  For example, LEMUR (GPT-4) solved 26 out of 50 SV-COMP benchmarks, while LEMUR (GPT-3) solved only 15. The average time per benchmark is also higher for GPT-3.5 Turbo, as well as the number of calls to an oracle.

These results demonstrate that the better an LLM model available as an oracle, the more performance gain it can provide to program verifiers like ESBMC and UAutomizer. **As LLM models are constantly improving, we believe that exploring how they can enhance the performance of these tools is very important. We believe our work is a significant step towards this goal.**

---

### Meta-Review · Area_Chair_xHuY · 2023-12-10

**Metareview:**

The reviewers were split about this paper and did not come to a consensus: on one hand they appreciated the clarity of the writing and the strong experimental results; on the other they had doubts about (1) the overall contribution, (2) the restriction on program length, (3) the benchmarks chosen, and (4) the time comparison. After going through the paper and the discussion I have decided to vote to accept because the authors respond convincingly to each of the main concerns. Specifically for (1) reviewers argued that the rules and algorithm 1 aren’t a strong contribution as the soundness proofs are simple and it seems to ignore the harder part of program verification: completeness. In a lengthy back-and-forth discussion the authors argued that proving the soundness of the rules was useful because it allows one to cover all possible proof derivations, not just the one used to create Algorithm 1. If someone were to derive an alternative algorithm using these rules the algorithm would already be sound by construction. Fruther they argue that there are other important works that other key verification works such as Model-Constructing Satisfiability Calculus do not have completeness but is still very useful. I believe that this fully resolves the concern, I agree that completeness should not necessarily be required for acceptance, particularly given that utilizing LLMs for program verification is a largely unexplored area. For (2), the reviewers argued that the restriction on program length (given the restriction on prompt length in GPT4) is a serious downside. They argued that the authors should investigate how to first decompose programs then apply their technique to each decomposed part. The authors responded that such decomposition is non-trivial as it would require the right specification language. I think a stronger response is that there is no fundamental restriction on LLM prompt lengths. Certainly it is a function of time and computing power, but an LLM could in principle be trained to accommodate any of the benchmark programs. For this reason I do not see this point as an issue. For (3) the reviewers argued that the authors may by cherry-picking benchmarks to suit their method. The authors pointed out that they specifically selected programs that current verification methods cannot solve, which resolves the point. For (4), the reviewers argued that the time comparisons where unfair because they do not count LLM time. There was a long back-and-forth on this point. The complaint evolves from requiring the same time limit for all methods to requiring the same amount of money (for the baselines, use the same amount of money on cloud computing cores as used by Lemur on GPT4). To the time point: I believe the authors already limit all methods to have the same time budget, including LLM calls. (@authors: if this is not the case please fix the timing results to include the full runtime including LLM calls). To the money point, this is out of scope as there are plenty of open source LLMs available to circumvent calls to. The authors even try running GPT3.5 and show that the proposed method is not tied to the performance of GPT4, as it still improves over prior work. Overall, the authors have convinced me that they are able to fix these concerns in the final version in their responses to reviewers. For these reasons I argue for acceptance. Authors: please make the changes recommended by reviewers, as you have already started to do. Once this is done, the paper will make a great contribution to the conference!

**Justification For Why Not Higher Score:**

The authors responded to all reviewer concerns but sometimes seemed to miss the main point. This prevented them from improving the paper further to reach spotlight / oral level. Overall, the paper is almost better fit for a PL conference, there is not a large ML contribution, but given the interest of the ML community in verification it seems many people would find this applied paper useful.

**Justification For Why Not Lower Score:**

The contribution is interesting and may enable further similar ML-augmented verifiers.

---

### Decision · Program_Chairs · 2024-01-16

Accept (poster)